# Communicating the Turkish Military Strength and Organisation after the Crusader Defeat at Nicopolis: Comparing Philippe de Mézières's *Une Epistre lamentable*, Honorat Bovet's *L'Apparicion Maistre Jehan de Meun* and Bertrandon de la Broquière's *Le Voyage d'Outremer*

**Zeynep Kocabıyıkoğlu Çecen**

Royal Holloway, University of London, Egham TW20 0EX, UK; zeynep.kocabiyikoglucecen.2021@live.rhul.ac.uk

**Abstract:** The outcome of the Nicopolis crusade (1396)—a crushing victory for the Ottoman Turks over a crusader army led by Franco-Burgundian knights—had particularly deep resonances in the Kingdom of France. This was reflected in several contemporary works that lamented and/or criticised the crusaders' defeat. Among these, Philippe de Mézières's *Une Epistre lamentable* (1397) and Honorat Bovet's *L'Apparicion Maistre Jehan de Meun* (1398) are of special note because they contain not only remarkable reflections on the campaign but also interesting observations on the successful Ottomans. Their praise of the Turks, especially regarding their military organisation and discipline, served both to criticise the crusaders' own lack of discipline but also to present an example for them to follow in order to avenge the defeat. As Nicopolis marked the beginning of a Burgundian claim to champion the crusading movement, these works were primarily addressed to the Duke of Burgundy, among other European princes and nobility. In the mid-fifteenth century, Duke Philip the Good would carry this claim to its zenith by several undertakings that included sending envoys to the East to gather information about the enemy. One of these envoys, Bertrandon de la Broquière, visited the Ottoman lands in 1432–1433 and some twenty years later wrote his *Le Voyage d'Outremer* to communicate the observations and intelligence gathered during his journey. The fact that this work, written almost six decades after Nicopolis, contains multiple allusions to the defeat, in addition to similar portrayals and comments on the Turks within the two earlier works, can be taken to suggest a continuity in Western military portrayals of the Turks from Nicopolis onwards.

**Keywords:** Nicopolis; Ottoman Turks; military discipline; knighthood; crusades; Burgundy

## 1. Introduction

We can concur that the Ottoman Turks had rightfully established themselves as the ultimate threat to Europe in the late fifteenth and sixteenth centuries. Such danger was manifest in their conquest of most of Eastern Europe coupled with advances towards Western Europe that threatened Italy and later saw the siege of Vienna. In parallel with this military activity emerged a rich body of Renaissance literature on the Turks, ranging in tone from demonising to laudatory and sometimes both at the same time (Lewis 1993, p. 74; Hankins 1995; Bisaha 2004; Schwoebel 1969). Whereas 'Turks' would gradually replace 'Saracens' in references to the 'other' towards the end of the fifteenth century, 1396 was an early date for them to be recognised, in Richard Knolles's words, as 'the terror of the world' if not for one fateful expedition: the crusade of Nicopolis, in modern Bulgaria (Kaçar and Dumolyn 2013, pp. 914–15, 920–21; Lewis 1993, p. 72). Described by historians as 'the first major encounter between the centralising Western European polities of the later Middle Ages and advancing Ottoman power', it brought a crusader army of the joint forces of the king of Hungary and Western warriors led by the son of the Duke of Burgundy, up

against the army of Sultan Bayezid I, resulting in a resounding Turkish victory (Kaçar and Dumolyn 2013, pp. 905–34; Gaucher 1996; DeVries 1999; Atiya 1934; Nicolle 1999).

From a certain point of view, the failure of the Nicopolis expedition might not have had a serious impact in Eastern Europe as the status quo the Balkan principalities had vis-à-vis the Ottomans was not really disturbed (Vaughan 1954, p. 24). On the Western side, while it was perhaps 'the most important crusade organised against the Turks in the fourteenth century', it was also 'the most severe crusading defeat since the days of St. Louis' (Housley 1992, p. 73; Morand Métiever 2018). Yet it is also debatable if, as Atiya argues, its effects in Western Europe were to 'discourage' fighting the Turks and to prompt an urge to leave 'Eastern Europe to its fate' (Atiya 1938, p. 463). As we will see, the Burgundian dukes were not remotely deterred from crusading pursuits against the Ottomans to the point of making it their state ideology. However, the severity of the defeat can be illustrated by the deep resonances it produced in the West, especially in the Kingdom of France (of which Burgundy was a vassal but also a power in its own right), provoking a range of reactions from despair and lamentation to questioning the causes of the defeat and criticising the knights who led the crusading army. We can see these reflected in a variety of contemporary texts belonging to different genres, such as chronicles, chivalric biographies and didactic literature. Significantly, some of these texts included detailed portrayals of the Turks in which their military qualities come to the fore and often receive praise (Kaçar and Dumolyn 2013, pp. 917–19; Kocabıyıkoğlu Çeçen 2021). Whereas books on the history, customs, armies, government, etc., of the 'Turk' written by all kinds of visitors to the Ottoman lands, such as slaves, refugees, pilgrims, merchants, and diplomats, were quite common in Renaissance Europe (numbering forty-nine in the English language alone at the start of the seventeenth century), they were not at all abundant at the end of the fourteenth and the beginning of the fifteenth century (Lewis 1993, pp. 75–77). Two of the works considered here, *Une Epistre lamentable* and *L'Apparicion Maistre Jehan de Meun*, were written in the immediate aftermath of the defeat and are significant for their interest in the Turks and their practices even if they do not provide detailed presentations of the Turks. In contrast, a third work, *Le Voyage d'Outremer de Bertrandon de la Broquière*, is a noteworthy precursor of the detailed reports on the Turks that emerged throughout and after the Renaissance.

## 2. Works

### 2.1. Philippe de Mézières's Une Epistre lamentable, Addressée en 1397 à Philippe le Hardi, duc de Bourgogne sur la défaite de Nicopolis (1397)

Among several works which allude to the defeat and often draw conclusions from it, Philippe de Mézières's *Une Epistre lamentable, Addressée en 1397 à Philippe le Hardi, duc de Bourgogne sur la défaite de Nicopolis* (1397) deserves special mention because the author does not merely deal with the defeat in a chapter or a part of the work but devotes the whole volume to it (de Mézières 2008). This is not altogether surprising as Mézières (1327–1405), 'the old pilgrim' as he refers to himself in allusion to his previous work, *Le Songe du Vieil Pélerin*, was a former counselor to King Charles VI and a seasoned crusader knight who had spent a great deal of his lifetime in the Mediterranean fighting the Turks and the Saracens. Mézières's crusade experience began with the Smyrna crusade in 1346 and was followed by a series of other campaigns across the Mediterranean. This inspired him to outline a program to defeat the Turks in which the Order of the Passion of Jesus Christ (*L'Ordre de la Passion de Jésus Christ*), an organisation that he himself had founded, would play an important role. His later sojourn in Cyprus as chancellor to King Pierre I between 1360 and 1369 would provide him with the first opportunity to promote his plans for a crusade. While Pierre's Alexandria campaign of 1365, which culminated in the sack and the subsequent abandonment of the city, did not live up to Mézières's idea of a successful crusade, he would conceive his Order in 1367 and subsequently persevere in promoting both it and his proposed crusade. He was able to do this in his capacity as an advisor to the Dauphin of France, the future King Charles VI, and as a member of the *Marmousets*,

the ghost counselors to the king during the early years of his reign. With crusade having become his lifelong passion, Mézières rewrote the rules of his order at least three more times between 1367 and 1396 and recruited around eighty members from among the French and the English nobility. Yet, despite his constant endeavours, Mézières could neither come close to attracting the thousand adherents he envisaged nor mobilise a crusade (Tarnowski 2006; Ioarga 1896; Blumenfeld-Kosinski 2009, pp. 174–81; Magee 1998, pp. 371–76; Atiya 1934, pp. 26–32).

Having also promoted his crusade scheme in his *Epistre au roi Richart* (1395), a letter advocating peace between the crowns of England and France a year before Nicopolis, Mézières's bitter disappointment in the failure of the Nicopolis expedition can only be imagined (de Mézières 1975). While the crusade set off in response to Hungarian and Byzantine envoys coming to the French court to plead for help against the Ottomans, it would not be co-led by either the English and French kings as envisaged by Mézières, or by the Dukes of Lancaster and Orléans who had initially been enthusiastic about taking the cross. Instead, it was Jean de Nevers, the twenty-five-year-old son of Philip the Good of Burgundy, who took command. Nor would Mézières's disciplined and pious Order of the Passion steer the crusade according to the carefully planned details he had laid out. Instead, the expedition turned out to be a chivalric spectacle for the youthful nobility of France and Burgundy, who were eager to fight the infidels for glory and renown and thereby serve Duke Philip the Bold of Burgundy's ambitions to be the champion of crusade (Housley 1992, pp. 75–79; de Mézières 2008, pp. 47–52, 59–60; Schnerb 1996).

Thus, Mézières wrote the *Epistre Lamentable* in the immediate aftermath of Nicopolis, tormented by the notion that the defeat could have been avoided had his counsel been heeded (de Mézières 2008, p. 227). He presents the *Epistre* to the Duke of Burgundy but addresses it to 'all kings, princes, barons, knights and commons of Christianity in its substance'. This suggests that perhaps abandoning his initial scheme of Anglo-French leadership of a crusade, he now regarded the duke as its leader yet still sought attention from all other Christian princes. As its title suggested, this was a text of lamentation for the defeat of the crusader army at Nicopolis but with an agenda to avenge it through the plans Mézières had already constructed in his previous works. Though he largely finds fault with the Hungarians and the schismatic Christians, Mézières describes the main reasons for the defeat as the corruption of the crusader army across the four virtues of rule, discipline, obedience and justice. He then advises how the defeat should be avenged by another crusade led by Christian princes and the Order of the Passion (de Mézières 2008, pp. 121, 128–32). This would entail a very detailed organisation and a call for participation from all over Europe. Although this part of Mézières's narrative, as in his other works, is heavy with allegories and references to biblical and classical examples, his plan first to recover prisoners from Nicopolis and then to inflict a defeat on the Turks includes quite practical details, some of which apparently relied on the author's knowledge of the Turks and their habits. Then follows a short but detailed summary regarding the strength of the Ottoman Turks and the several rival principalities (*beyliks*) that had previously reigned in Anatolia, including a review of how Sultan Bayezid came to rule over both these and Christian lands (de Mézières 2008, pp. 210–17). This piece of information is of interest because it gave a broad picture of the Turkish landscape and correctly assessed Ottoman strength in comparison with its competitors. As he also proclaims himself, this is first-hand knowledge from his time in the service of the king of Cyprus when he gathered information through merchants and Saracen renegades (de Mézières 2008, p. 212). Whereas we can observe some confusion and error regarding geographical locations and the spelling of proper names (not at all uncommon in contemporary writings), overall, this is a good and perhaps rarely found assessment of who was who in fourteenth-century Anatolia. Moreover, here Mézières tends to stray away from his usual allegorical and sermon-like tone to an informative one, revealing the importance he attached to having an understanding of the Turks prior to taking any action against them, or as he declares, 'knowing the enemy and his conditions may be said to be half the victory' (de Mézières 2008, p. 210). He might also be alluding to the Nicopolis crusaders

who allegedly chose to ignore the Turks' military status and strategies before and during the battle, thereby causing the defeat. Contemporary chroniclers Froissart and the Monk of Saint-Denis reveal that the crusaders, who were seated at lunch when they heard the news of Bayezid's approach near the crusader camp, were taken by surprise and hurriedly tried to prepare, allegedly 'with wine in their heads'. It is also suggested that even though scouts had earlier brought news of the Turkish approach, the young leaders of the crusade, the marshal and the constable of France, Bouciquaut and the count of Eu did not hear them. Moreover, these sources also condemn them for their presumption in stubbornly refusing to listen to the king of Hungary, who, entirely familiar with Turkish tactics, wisely suggested putting his irregulars in the vanguard instead of the Franco-Burgundian knights to break the enemy forces. In response to these criticisms, the anonymous author of the biography of the Marshal, *Le Livre de fais du bon Messire Jehan le Maingre, dit Bouciquaut*, goes to great pains to defend his hero (Bellaguet 1840, pp. 499–505; Froissart 1871, pp. 311–15; Lalande 1985, pp. 102–4).

Whereas Mézières makes several disparaging references to the Turks throughout his narrative, such as 'vile', 'savage', 'fierce men, cruel and ill-taught', and 'true enemies of the faith' who are also suspect in the sincerity of their own belief, he also praises them for their military valour and discipline (de Mézières 2008, pp. 155, 157, 172, 185, 227). First of all, he acknowledges that Amourath (Sultan Murad, r. 1362–89) came to rule over other principalities in Anatolia by virtue of not only 'treaties and tyrannical conquests' but also his valour, and also died valiantly in a war against Walachia (Battle of Kosova, 1389) (de Mézières 2008, pp. 215–16). He also establishes that,

> the aforesaid Amourath and his son Baxeth have not conquered the aforesaid empires and realms so easily that one can think' but 'they have conquered them by valour of arms and by order well-guarded in their army . . . (de Mézières 2008, p. 216)

This was not the first time Mézières recognised the military valour of the Ottomans, for in his *Songe du Vieil Pélerin* (1389), he had already described the Turks as 'robust and very valiant combatants' as well as 'very brave' (Pippidi 2013, p. 12; de Mézières 1969, I, p. 253). Moreover, because the strength of the Turkish army came from its military discipline, the army to defeat them should also be disciplined. Mézières asserts in *Epistre* that,

> We can sufficiently recognise the power of this aforesaid Baxeth (Bayezid) and his knighthood and the experience of his army well-guarded by the discipline of rules of chivalry. And for this, it is of pure necessity that the men of arms that take up the war against these Turks should be well-regulated in God and continuously guarded and comforted by sweet Christ, their patron. (de Mézières 2008, p. 184)

He then offers a detailed narrative of the organisation of this army, including the mobilisation of troops from across Europe under the command of the heads of the Order of the Passion, and also indicates the routes they should follow (Philippe de Mézières 2008, pp. 184–92). This elaborate plan outlining the most effective combat strategies against the Turks once again falls back on his own experiences of crusading and knowledge of the Turks, as well as on the experiences of previous crusaders.

Even though Nicopolis was only a brief encounter between Western Christendom and the Turks, so disastrous were the consequences that Mézières was promoted to write that 'Bayezid previously did not go so far in being victorious against Catholic Christianity as the present day' (de Mézières 2008, p. 187). He further warned that it was not only the Kingdom of Hungary, which if not helped by the Order of Passion and other Christian forces, 'is in great peril', but also the rest of Christendom, although they may not 'feel assaulted and injured by the horrible wound that is repeated so many times', who will foolishly say that 'the Turks are not yet at the Charenton bridge' near Paris (de Mézières 2008, pp. 216, 219).

Although Mézières's long and verbose lamentation after Nicopolis only comprises a brief summary of how the Ottomans came to rule over Anatolia and established their military power, as well as his suggestions as to how to defeat them, it still provides an interesting example of an early assessment of the Ottoman Turks at a time when they were not a household name in France or across the rest of Western Europe. Moreover, it is also remarkable to find 'an old pilgrim' like Mézières praising enemies of Christ for their valour and military discipline and acknowledging their conquests as well-earned. Even though *Epistre Lamentable* was clearly destined to stir its audience to take up arms in retaliation for Nicopolis, Mézières did not simply confine himself to why the faithful faced such a defeat or the perils awaiting Christendom if the infidels were not driven away, but he also made an assessment of the military strength and organisation of the Turks so that knowledge of the enemy would help defeat them. In this, it is noteworthy to see him give credit to their military discipline and to register the stark contrast he draws to the absence of such a basic military attribute in the crusader army at Nicopolis.

Mézières's plans to assemble an international order of pious knights, led by the Western monarchs, to defeat the Turks were never realised either due to the continuous wars between the kings of England and France, or to his own failure to exert influence on the crowned heads of Europe because he was cast out from the political scene of France after the early 1390s (de Mézières 2008, p. 212). Yet his observations on the Turkish military and political power are certainly worth attention because not only they derive from first-hand information, but also, they provide an early Western viewpoint of Turks.

### 2.2. Honorat Bovet's L'Apparicion Maistre Jehan de Meun (1398)

Another interesting text contemporary with *Epistre lamentable* was Honorat Bovet's *L'Apparicion Maistre Jehan de Meun* (1398). Although not directly about Nicopolis itself, it contains several allusions to the defeat accompanied by interesting remarks about the Turks (Bovet 2005). This long poem is dated 1398, but given its remarks about Nicopolis prisoners, it was probably written sometime before August 1397, when they had been released. Bovet was a Benedictine monk and canon lawyer from Provence who has left us only two other surviving works, namely *L'Arbre des batailles* (1387) and *Somnium super materia scismatis* (1394) on the laws of warfare and the Great Schism in the Catholic Church (1378–1414) respectively. *Apparicion* is Bovet's last known work and includes his thoughts both on warfare and the Church, among other contemporary controversies. It is a satire written in the dream vision genre and dedicated primarily to the Duke and Duchess of Orléans and then to the Duke of Burgundy, two contestants for the rule of France at the time, with the latter also being the instigator of the Nicopolis crusade. Bovet's seeming aim in this work and others, just like that of Mézières, was to highlight contemporary ills, including the failures of knighthood, while proposing solutions to them (Batany 1982, pp. 21–30; Bovet 2005, pp. 4–15). While the author's choice of the Nicopolis defeat to criticise contemporary French knighthood is quite significant for showing the impact of the defeat in the Kingdom of France, his use of an imaginary Saracen (Turk) envoy from the court of Bayezid to communicate this criticism is also striking because it reflects both Bovet's view of the Turks especially concerning their military status and also his presumptions about how the Turks might view the Christians.

Bovet's comments on the Turks, just like that of Mézières, focus mainly on their military discipline, which also serves to sharpen his criticism of the French for their failings. Thus, the Saracen's main reproach for the French knights, namely their love of luxury and comfort, is pitted against the Saracen (Turkish) warriors' simple lives, and this comes with an allusion to the Nicopolis defeat when the Saracen declares,

> . . . we Saracens, on the other hand,
> As my lord of Nevers knows,
> We live otherwise, for certain:
> Clear water and a bit of bread
> Is a big meal for a Saracen,

> So there's no worry over cellared wines,
> Or what meat is in season;
> If any is found, that is first-rate.
> And when it is time to go to bed,
> He does not worry about disrobing,
> Or trouble himself with looking for straw,
> But only with finding some solid ground. (Bovet 2005, pp. 89–91, ll. 435–46)

Another comparison he makes is between the French and the Turkish endurance in battle. The Saracen observes that,

> Your armour is too heavy,
> Which means that when you are fully armed,
> In a short time, you are crushed;
> . . .
> And if a man in armour falls,
> He'll be slow in raising his head,
> . . .
> The Saracens arms themselves lightly,
> . . .
> And can endure a long time in battle;
> And, of course, they ride very well
> For they have developed great stamina,
> And in battle, they willingly endure for long periods
> Pain, fatigue, heat and cold. (Bovet 2005, p. 95, ll. 524–41)

Still, Bovet asserts that if the knights cared as little about luxury and comfort as the 'Saracens' and 'obeyed their own laws', there would be no one who could stand against the French (Bovet 2005, p. 101, ll. 671–76). Still, perhaps assuming that it would not be realistic to expect this from the noble knights, he comes up with an alternative crusade scheme like that of Mézières. While Bovet's scheme does not involve a detailed description of the routes to follow or the strategies to be adopted, or a specific Order such as Mézières', it involves a quite radical, namely the suggestion that an army be recruited from among peasant workers,

> Can serve effectively as soldiers.
> . . .
> Because, as a result of their austere peasant experience
> They fear neither bad bedding nor bad bread
> Nor wind nor rain nor prolonged hunger;
> And they can bear any exertion,
> And have the practiced arms for delivering
> Heavy blows, and for great tenacity,
> Because they are accustomed to enduring pain.
> . . .
> And also, less is lost
> Should they die in battle;
> And if they land in prison,
> Christendom will not suffer such a setback,
> Nor such shame as the nobles. (Bovet 2005, p. 97, ll. 565–78)

While Bovet's idea of a crusader army of peasant workers seems quite radical for fourteenth-century France, he may well have been influenced by the disciplined Turkish warriors, few of whom were noblemen—unlike the knighthood of France—and thus accustomed to hardship and being content with little. Although the first standing armies in France were yet to be assembled (this took place around the middle of the fifteenth century), it is quite interesting to find a blueprint proposed by a late fourteenth-century monk inspired

by Turks via the Nicopolis defeat. Around the same time as Bovet, the Chancellor of the Florentine Republic, Coluccio Salutati, was making very similar remarks about the Turkish *janissaries*, the elite Ottoman corps who were recruited from among Christians, praising them for their rigorous training and discipline, which enabled them to endure all hardships (Bisaha 2004). Whereas Bovet did not have the same exposure to the Turks as Mézières did or live closer to the Turkish threat as Salutati did, it is highly probable that he received information about the Turks at the noble courts he served. Thus, he could have come across survivors of Nicopolis, such as Jacques de Heilly (who was sent as an envoy by Bayezid after the defeat) or others who escaped the battlefield or at least heard from noblemen and knights who have met them. Moreover, the comments of the Saracen character and allusions to Nicopolis crusaders, together with his criticisms of knights, echo those made by chroniclers and other contemporary writers after the defeat and are evidence that it was not only the knightly classes or crusade enthusiasts, such as Mézières, who were concerned by the failure of the Nicopolis defeat (Bellaguet 1840, pp. 498–99; Deschamps 1934). While Bovet's was not a wholesale examination of the defeat as that of Mézières, his indirect address of its reasons through a Saracen agent leaves a more dramatic impression of the impact of Nicopolis in France.

### 2.3. *Le Voyage d'Outremer de Bertrandon de la Broquière*

Both Paviot and Kaçar-Dumolyn consider Nicopolis to be a prelude to the Burgundian tradition of crusading, which was largely shaped during the rule of Duke Philippe the Good (1396–1467). While the duke, born just a few months before the disastrous expedition led by his father Jean the Fearless, might or might not have grown up with stories of the fearful Turks, his dukedom was certainly a time that Burgundy showed great enthusiasm towards crusading and employed it as a state ideology. Whereas the Burgundian ideologues also fell back on other crusading ancestors, such as Godfrey de Bouillon and Baldwin IX of Flanders, to style Philippe as *athleta Christi*, it is also reasonable to look back to his grandfather, Philippe the Bold, for this role. As Philippe the senior had been ambitious enough to try to fill in the shoes of the French kings by being the leading figure in the crusade of Nicopolis, his grandson clearly took up the mantle of crusade champion, motivated more by political ambitions and commercial interests in the East (of Flanders that was then under Burgundian lordship) than piety (Atiya 1934, pp. 39–41; Paviot 2004, pp. 70–71; Kaçar and Dumolyn 2013, pp. 913–19; Housley 1992, pp. 78–79; Finot 1890, p. 8).

Although Philippe the Good never went on a crusade, except for two failed attempts in 1454 and 1463, his manoeuvring for the role of *athleta Christi* began very early on. Evidence for this can be found in his foundation of the Order of the Golden Fleece (*L'Ordre de la Toison d'Or*) in 1430, his annual donations to Christian communities in Jerusalem at least from 1435 on, his sending of relief forces on several occasions against the Mamluks in 1429, 1441 and 1444 and against the Ottomans in 1444–1445 (although this lasted up until 1448, in practice, it turned into piracy while fighting the infidels on the Barbarian Coast, Rhodes, Morea and the Black Sea). We can also add his dispatch of two different envoys to the East within a decade to gather information about the Saracens and the Turks, namely Ghillebert de Lannoy and Bertrandon de la Broquière, respectively, in 1421 and 1432, and his embassies to mobilise a crusade sent to the several European courts, including those of France, England, Holy Roman Empire and the pope in 1451. His first attempt to go on a crusade, in response to Pope Nicholas V's bull in March 1454, began with his organisation of the 'Feast of the Vow of the Pheasant' (*Banquet de Voeu de Pheasant*), where the participants swore a variety of oaths to avenge their defeat against the Turks. Although this looks more like pomp and play, the fact that the duke took crusading plans seriously can further be demonstrated by two separate memoirs on how a potential crusade against the Turks should be conducted, presented to him by the Burgundian leaders of the 1444 crusading fleet and his counselors, respectively, in 1456 and 1457. It was also around this time that he asked Bertrandon de la Broquière to write about his 'secret' travel to the East. Although internal conflicts constantly delayed Philippe's ability to realise his crusade plans, he made a last attempt in 1463–1464

to dispatch troops to join those of Pope Pius II, who had gallantly offered to personally lead the crusade he had been preaching for about a decade. The final blow to this final abortive attempt was the death of Pius himself (Finot 1890; Schwoebel 1963; Paviot 2004, pp. 70–80; Paviot 2018, pp. 135–38). Yet, the fact that Philippe the Good was directly approached by the Byzantine Emperor John Paleologus for help against the Turkish siege of Constantinople in 1442 and Pope Nicolas V for a crusade against the Turks after their capture of Constantinople is evidence that the duke's efforts had produced the intended effect (de la Broquière 1892, p. xvii; Finot 1890, pp. 6–8; Kaçar and Dumolyn 2013, p. 910). Whereas neither Charles VII nor his successor Louis XI granted Phillip permission to go on crusade, the duke's enthusiasm probably raised papal hopes for mobilising a crusade with Burgundian leadership. The intended crusade would finally take off the following year during the pontificate of Nicolas's successor, Calixtus III, but without Burgundian participation (Paviot 2004, pp. 74, 78).

As noted above, the author of *Le Voyage d'Outremer de Bertrandon de la Broquière*, the first esquire to the duke, was sent on a mission to the Outremer in 1432 after his first envoy Ghillebert de Lannoy, dispatched in collaboration with Henry V of England in 1421, had failed to reach Turkish lands because of the civil war between contestants for the Ottoman throne. While Broquière describes his own journey as an ordinary pilgrimage which was diverted to Anatolia when he decided to take the land route back home, he also reveals that his book should be a guide to those attempting to conquer Jerusalem. However, despite these statements, he does not really elaborate on his journey to Jerusalem on the grounds that the road to the holy city is already known to most people, and he goes on to describe the Turks, their land and customs (de la Broquière 1892, p. 2). While most of Broquière's travelogue comprises a detailed report on the land of the Turks, in a separate section towards the end, he elaborates on their military practices and makes projections for a potential army that would defeat them. It is significant that Broquière's travel notes comprising such a report were presented to the duke around the same time as the other aforesaid memoirs on how to conduct a crusade against the Turks. They all attest to how serious Philippe the Good was in his planning of the crusade and, unlike his father, valued information from those knowledgeable about the Turks (Finot 1890, pp. 11–20).

Despite his apparent recognition of the Turkish threat and the real purpose of his own mission, it is significant that Broquière still envisions the eventual goal of the prospective crusade after defeating the Turks as Jerusalem (de la Broquière 1892, p. 230). The fact that the same end goal was true for both the Nicopolis expedition and Mézières's planned crusade may demonstrate that reconquering Jerusalem was still perceived as the ultimate achievement of any crusade expedition at this time, with a *passagium generale* following a *passagium particulare* (Froissart 1871, p. 220; Housley 1992, p. 78). Mézières, always the idealist, had even frowned upon the Nicopolis crusaders going for the Balkans instead of the Holy Land as 'a perversion of the true goal of French chivalry' (Blumenfeld-Kosinski 2009, p. 182). Housley discusses the reality of the goal of Jerusalem persisting in the fourteenth century and beyond. He concludes that although the urgency of the Turkish threat was widely acknowledged, it was the memory of heroic predecessors and the necessity of appealing to traditions of crusading that prompted appending the idea of the recovery of the Holy Land to calls for a new crusade against the Turks (Housley 1992, pp. 46–48). Although ousting the Turk from Europe, communicated in Pope Eugenius IV's call for a crusade in 1444, would be a catchphrase for future crusades against the Ottomans, the presence of a Turkish threat in wider Europe had probably not yet been established before the Ottoman conquests in Italy and Hungary. Thus, fighting the Turk might not have resonated with Western crusaders at large, excepting Cypriots, Venetians and Hospitallers who were constantly waging war against them in the Mediterranean (Paviot 2018, p. 136). Vaughan argues that it was only after 1453 that the destination of the Holy Land and Jerusalem in crusade discourses began to be replaced by Greece and Constantinople, accompanied by calls for the expulsion of the barbaric Turks from Europe (Vaughan 1954, p. 66).

Yet the memory of fighting the Turks at Nicopolis seems to be quite alive in Broquière's Burgundy as his travel notes suggest. His narrative of the military practices and tactics of the Turks comprises quite a few remarks explaining the defeat, which were probably quickly picked up by at least his knightly audience, a group who would benefit from his advice in future crusade planning. The most direct of these comments is the reference to the defeat of 'the emperor Sigismund and the duke Jean' when Broquière mentions how the Turks always triumphed over the Christians in battle (de la Broquière 1892, p. 222). He then gives Nicopolis as an example of why the crusaders should take advice from those familiar with the Turkish way of fighting. He argues that according to what he has heard from his sources, if 'in the recent past the crusaders wanted to believe the emperor Sigismund, the emperor would not have needed to leave the battleground' (de la Broquière 1892, p. 225). This, as has been mentioned above, is another allusion to the young leaders of the Nicopolis crusade not heeding Sigismund, who advised them to wait in the rearguard. This is also evidence that even after more than half a century, the criticism of Nicopolis crusaders for being defeated due to their stubbornness and pride still persisted. In Broquière's narrative, we can also find some indirect references to Nicopolis. For example, his stipulation that a prospective crusade should not be undertaken for renown and glory but for the grace of God, although a commonplace in clerical crusade sermons, can be read as an explanation to why Nicopolis was such a disaster, with the portrayals of the crusaders in the accounts of the campaign in mind. Although it was a criticism in some portrayals of the campaign, such as 'Faicte pour ceuls de France quant ils furent en Hongrie' of the contemporary poet Eustache Deschamps who deplored the expedition as a chivalric spectacle to earn glory and renown, in others, like Marshal Bouciquaut's biography, it just described the mood of the crusaders (de la Broquière 1892, p. 225; Atiya 1934, p. 131; Lalande 1985, p. 88). Other allusions to Nicopolis may be Broquière's mention of the Turkish practices of setting an ambush and pretending to pursue troops in order to disband them, which were exactly the Turks' successful tactics at Nicopolis, also told in the chronicles (de la Broquière 1892, p. 222; Froissart 1871, p. 315). Likewise, the Turkish armies' moving swiftly and quietly from one place to another recalls Bayezid's aforementioned sudden appearance at Nicopolis, when he was thought to be in Anatolia (de la Broquière 1892, pp. 220–21). Thus, Broquière, with these constant reminders of Nicopolis, seems to warn against making the same mistakes in a forthcoming crusade.

These and other information about the military customs and tactics of the Turks in *Le Voyage d'Outremer* are communicated for the purpose of showing 'the ways to break and defeat them in battle and with which men and conquer their dominions' (de la Broquière 1892, pp. 216–17). These, either the author's direct observations or those conveyed to him by eyewitnesses, almost exactly fit the portrayals and comments made earlier by Mézières or Bovet. Although Broquière refutes the common saying 'strong as a Turk' on the grounds that there are stronger Christians and weaker Turks, he nevertheless gives credit to the Turkish soldiers for their diligence and discipline, describing them as 'living on little, just as a piece of poorly baked bread and raw meat that is little dried under the sun', etc., and 'sleeping on the ground' (de la Broquière 1892, p. 217). These details highlight the Turks' disciplined way of life and are very much reminiscent of Bovet's descriptions of Turks quoted above.

Likewise, Broquière's praise of the Turks, as 'honest and obedient men . . . who have come so far by their valour' and 'made the great conquests which surpass the Kingdom of France in greatness' and defeated the Nicopolis army sound like an echo of what has been previously quoted from Mézières's comments about the conquests of Murat I and Bayezid (de la Broquière 1892, pp. 221–22, 224). Interestingly enough, Broquière's high opinion of the Turks is totally in contrast with his dislike and distrust of the Greeks as he openly declares that 'he trusts the Turks more and has found more friendships among them' (de la Broquière 1892, p. 149). This attitude, however surprising, is not so different from that of a contemporary traveler, the Spaniard Pero Tafur, who visited the Ottoman court during his travels to the Holy Land and Constantinople in 1437–1438. In addition

to his admiration of the Turkish armies and their remarkable endurance (something that also accords with the views of the authors discussed here), Pero echoes Broquière in his description of the Turks as 'noble, truthful, merry and benevolent', a complete contrast to the 'unreliable and vicious' Greeks (Rodriguez 2015, pp. 316–18; Pippidi 2013, p. 13). Moreover, Broquière gives very detailed descriptions of the Turkish soldiers, including their faces, their physiques, clothes, horses, the way they ride, etc. (de la Broquière 1892, pp. 216–21). Yet, what he underlines regarding the Turks is the same as Bovet and Mézières: their military discipline. From the way they assemble their armies, how they take off and ride all night without stopping, to how they follow their enemies to eventually disband them, the Turks are described as an embodiment of order and discipline. Moreover, their obedience to their lords is such that they do not dare to transgress for fear of their lives (de la Broquière 1892, pp. 221–23).

Again, as in Bovet and Mézières, Broquière asserts that these disciplined Turkish troops can only be defeated with an organised and well-commanded army, which can stand firmly together against the great numbers of Turks and not be disbanded (de la Broquière 1892, pp. 224–25). 'Otherwise', he warns, 'whoever comes to the country (of the Turks) will be annihilated by them' (de la Broquière 1892, p. 228). In addition to the aforementioned Turkish strategies that the crusaders should be aware of and the lessons of Nicopolis that must be learned, he—like Mézières—also defines the structure of the army to defeat the Turks. Although not directed by a religious order, Broquière's crusader army also has an international structure, albeit confined to English, French and German troops. He gives a detailed description of these well-organised and firmly commanded troops, including the number of men and the arms and armour they should be using. It is quite noteworthy that he is meticulous enough to compare the notches in Turkish bows with those of Western bows and concludes that even though the Turkish bows might be stronger, the Turkish arrows are not as strong as those of the Christians, thereby rendering unnecessary the crusaders' need to wear thick armour (de la Broquière 1892, pp. 225–30). Among these details, the Turkish horsemen enduring long journeys and the need for the crusaders to be light-armoured against the light-armoured Turks echoes Bovet's comments about the Turks being good riders fighting with lighter armour and greater stamina than the French in heavy armour (de la Broquière 1892, pp. 217–19, 226–28).

Thus, the journey taken by Bertrandon de la Broquière about thirty-five years after Nicopolis can be seen as a response to correct what went wrong by collecting information about the Turkish military organisation, commissioned by the successor to the leader of the said expedition. The fact that his eventual presentation of a written account of his travels to the duke was probably triggered by another Christian defeat at the hands of the Turks, the conquest of Constantinople, reinforces the possibility that his observations and advice were supposed to be used in a forthcoming expedition by his patron. Broquière's detailed narrative, a product of these excellent observations about the Turks, reveals his mission to have been a real success, and his allusions to Nicopolis provide evidence that he established the link between the mission and the unfortunate defeat that had such disastrous results for Burgundy and France.

### 3. Conclusions

Whether *Le Voyage d'outremer de Bertrandon de la Broquière* could have been penned to prepare for a new crusade against the Turks after the conquest of Constantinople, prior to 1396 the Turkish threat to Europe had not yet been established. The Nicopolis crusade might have marked the beginning of the Western awareness that the Turks would be the great enemy of Christendom because the disastrous outcome of the battle generated a major backlash, particularly in France, which had contributed by the greatest number of knights. Philippe de Mézières's *Une Epistre lamentable* and Honorat de Bovet's *L'Apparicion Maistre Jehan de Meun* are among other works which gave long or short accounts and critiques of the defeat. They can both be cited as examples of contemporary reactions to the defeat,

which contained remarkable portrayals of Turks that conveyed both awe and fear (Kaçar and Dumolyn 2013, p. 915).

When we compare the observations and comments about the Ottoman Turks in these earlier French texts, one written with first-hand knowledge of the Turks, the other not, with *Le Voyage d'Outremer*, which was written more than half a century later, based on both the author's direct observations and his collection of information, we can note striking similarities. Among other shared details, such as the armour, arms, horses and military tactics of the Turks, the most important common emphasis regarding the Turks in all these works is certainly their military discipline. While the authors are all genuinely concerned about the Turkish threat, they also agree that if the French armies (or Western forces in general) had been as disciplined as the Turks, the latter would not have been able to defeat them. Whereas Western praise of military power and discipline of the 'other' was in no way unprecedented for we can find allusions to 'strong, brave and skillful' and highly disciplined enemy in earlier sources, such as the first crusade chronicles *Gesta Francorum* and *Gesta dei per Francos*, and later in the Western accounts of the Mongols, in the aftermath of Nicopolis, these remarks were not solely meant to establish the 'other', in this case the Ottomans, as the true enemy but also to inspire and draft a military organisation to defeat them (Hill 1967, p. 21; Levine 1997, p. 68; Jackson 2016, p. 73; John of Plano Carpini 1955, pp. 46–48).

While each author advocated his own version of this disciplined army, it is noteworthy that all versions drew on their understandings of Turkish military power and organisation. We can, therefore, speculate that their observations might have contributed to the question of the effectiveness of medieval armies led by knights and their gradual replacement by standing armies in France from the middle of the fifteenth century onwards. On another note, the similar outlook and observations on the Turks in the earlier texts and *Le Voyage d'Outremer*, as well as the latter text's direct and indirect allusions to Nicopolis, are indicative of a continuity running from Nicopolis to the late fifteenth-century in the context of Burgundy's adoption of crusading as its state ideology.

**Funding:** This research received no external funding.

**Institutional Review Board Statement:** Not applicable.

**Informed Consent Statement:** Not applicable.

**Data Availability Statement:** Not applicable.

**Conflicts of Interest:** The author declares no conflict of interest.

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
