# Peer review of "Communicating the Turkish Military Strength and Organisation after the Crusader Defeat at Nicopolis: Comparing Philippe de Mézières’s Une Epistre lamentable, Honorat Bovet’s L’Apparicion Maistre Jehan de Meun and Bertrandon de la Broquière’s Le Voyage d’Outremer"

_religions, doi:10.3390/rel14111386_

Round 1
Reviewer 1 Report
Comments and Suggestions for Authors
Review of “Communicating the Turkish Military Strength and Organization …”
I really like this article. The author has taken a subject that has been of recent scholastic interest and has presented a fresh look with some very valid and interesting conclusions built on solid research derived from original sources. His/her look at three contemporary or near contemporary sources on or influenced by the defeat of western European forces at the battle of Nicopolis is substantial. I have a few suggestions, although none of them concern me enough to encourage the publication of this article:
ll. 131-134: the author has rather simplistically summarized what is clear from the sources he references to as a more complex meeting between western European leaders and the king of Hungary whose greater experience in fighting the Turks was ignored.
ll. 138-139: “military valour” needs definition here
l. 217: “Saracens” should be plural
l. 251: “end of the day” is an inappropriate cliché
ll. 255-259: This sentence is very confusing
ll. 293 on: For some reason the author refers to the first Valois duke of Burgundy as Philip the Bold but the third as Philippe the Good. If the name Philip is to be Anglicized it needs to be consistently so. If the name is to remain French it needs to be Philippe le Hardi and Philippe le Bon. The two dukes are known primarily by their Anglicized names, but consistency is the most important here6
l. 301: Baldwin of Flanders should be Baldwin IX of Flanders
ll. 306-307: “taking the cross” does not always mean going on Crusade but simply preaching it and funding it – both of which Philip did in the years the author indicates
l. 314: “to avenge the Turks” means to fight on behalf of the Turks not on behalf of their enemy
ll. 316-317: Philip’s aid to the Knights Hospitallers at Rhodes was against the Ottoman Turks; the Mamluks did not extend their reach as far as Rhodes
l. 351: there is a typo in this line
l. 386: Eustace Deschamps was indeed a poet, but not “court poet”
ll. 459-460: The author’s assertion that Nicopolis first made the western Europeans aware of the Ottoman Turks is a bit of an overstatement. I’m not sure it is accurate, especially as every crusade preached in the fourteenth century, including those of Philip VI, John II, and Charles V, targeted the Ottomans as the enemy. It is a statement that can stand although I would prefer something like “the battle of Nicopolis proved to western Europeans that the Ottoman Turks would be a formidable opponent”
Comments on the Quality of English LanguageAuthor Response
I have made all the necessary corrections he has suggested, I am just confused on one comment regarding Burgundian fleet not fighting the Mamluks as Jacques Paviot (2004) records that relief forces to have been sent to Rhodes against the Mamluks in 1429,1441 and 1444 and Jules Finot (1890) also asserts that the siege of Rhodes by the Mamluks sometime around 1442-44 was lifted by a fleet led by Geoffrey de Thoisy. Paviot (2018) confirms that Thoisy went to Rhodes before helping out the Varna crusade, thus arriving late there.
Reviewer 2 Report
Comments and Suggestions for Authors
The article is very brief, does not make any clear argument, and does not even systematically introduce the choice of three texts you are reading for the topic. Stressing repeatedly that a text is "interesting" does neither qualify the choice nor does it specify the "interest" for the topic at hands. Comapring Philippe and Honorat is feasible. But when you come to Bertrandon, things get difficult. Neiher do you explain what this text has to do with the two others, not is there a clear argument that it is something else.
The bibliography is very short; even literature directly connected to the topic is missing such as Morand Metivier, Charles-Louis, Creation and Union through Death and Massacre: the Crusade of Nicopolis and Philippe de Mézières' Epistre lamentable et consolatoire, in: Turner, Wendy J. / Lee, Christina (ed.), Trauma in medieval society, Leiden 2018, 298-319; Biu, Hélène, " Tancier et fièrement parler " Honorât Bovet et " Maistre Jehan ", in: Boudet, Jean-Patrice (Ed.), Jean de Meun et la culture médiévale: littérature, art, sciences et droit aux derniers siècles du Moyen Âge, Rennes 2017, 257-300 --- let alone Bertrandon de la Broquière whose work has be subject to many studies not the least lately. And this does not yet consider the vast "image of the Turk" literature, or the "image of the other" problematic. The Ottomans were certainly not the first to be taken as an example for Christian warriors (think only of Saladin and of the Mongols).
A lot of context is missing as well. You may want to stress that liberating Jerusalem remains a main goal for the crusades to the Levant - although this has rarely been doubted. But your argument gets queitte distorted when you claim that the goal of Jerusalem had "justified the diversion of the Fourth Crusade". This is wrong - what on earth would have been wrong with the 4th crusade going directly to Jerusalem, as planned? - and it is difficult to guess what you actually want to say. You can of course claim that the goal Jerusalem had no reality anymore in the 15th century, but given the many crusading plans until at least the 16th century, you should at least give a good reason for that claim.

Author Response
I hope to have made my argument clearer now. It was how the communications of the Turkish might were recorded by the French authors in the immediate aftermath of the battle of Nicopolis and how much Bertrand de la Broquiere’s observations on the Turks made by half a century later had in common with these recordings that did not rely on any first person experience of the Turks, Broquiere’s account attesting to the Burgundian claim on crusade championship starting with Nicopolis.
I have added some context regarding the Western Image of the Turks and made use of some other sources on Nicopolis and later crusades, and also tried to clarify what I meant by the relevance of the end goal of Jerusalem for the later crusades.
Of the recent sources recommended by thew reviewer, I have made use of Morand Metiever's but not the other one as it was a literary examination of Bovet's work, therefore not relevant to my discussion. Yet I have added other sources on Mezieres, Nicopolis and the Western Image of the Turks.
Round 2
Reviewer 2 Report
Comments and Suggestions for Authors
The additional work on the article made it much better; thank you for doing this work.